

# Estimating near-surface specific humidity over the ocean

Anna Lea Albright[1], Bjorn Stevens[2], and Martin Wirth[3]

[1]Harvard University Department of Earth and Planetary Sciences, Cambridge, MA, 02138, USA
[2]Max Planck Institute for Meteorology, 20255 Hamburg, Germany
[3]Institut für Physik der Atmosphäre, Deutsches Zentrum für Luft- und Raumfahrt (DLR), Oberpfaffenhofen, 82234 Wessling, Germany

**Correspondence:** Anna Lea Albright (annaleaalbright@fas.harvard.edu)

**Abstract.**

The surface latent heat flux is a large term in the surface energy balance and difficult to estimate remotely. The main difficulty for its estimation remotely is a poor ability to measure near-surface humidity. Current methods to retrieve near-surface specific humidity approach the problem statistically and have errors of approximately $1\,\mathrm{g\,kg^{-1}}$ even in global-annual averages. Using extensive measurements from the EUREC$^4$A field campaign (ElUcidating the RolE of Clouds, Circulation Coupling in Climate), we demonstrate that remote-sensing measurements of cloud base height can provide useful estimates of near-surface humidity. Applying the method to 171 coincident radiosonde and ceilometer pairings collected from a research vessel from January 18 to February 14, 2020 yields skillful predictions of near-surface specific humidity regarding the mean (mean bias $0.33\,\mathrm{g\,kg^{-1}}$ compared to observed) and its variability (r = 0.76). We next apply this method using an airborne lidar to estimate cloud base height from above. In two case studies, we find similar skill in the predicted humidity, with low mean biases ($-0.06$ and $-0.03\,\mathrm{g\,kg^{-1}}$ compared to observed) with substantial variability captured (r=0.61 and r=0.57, respectively). Two main error sources, (i) the relative humidity lapse rate below cloud base and (ii) the temperature difference between the sea surface and near-surface air, are identified and quantified. Our proposed approach allows for estimates of the near-surface specific humidity using downward-looking space-borne lidar. This proof of concept raises the potential for its global application and for improved observational constraints on the surface energy budget.

## 1 Introduction

The surface energy balance is a fundamental property of the climate system. How it is partitioned among its different components, and how it varies in space and time tempers the behavior of the atmosphere above, and the land or waters below (e.g., Boccaletti et al., 2004). Among its varied components, the main balance is between moisture fluxes, extracting energy from the surface through evaporation, and the intake of energy from the sun. Sensible energy transfers, and net radiant energy fluxes in the thermal infrared also combine to cool the surface, but on average only half as strongly as the evaporation of water which maintains the flux of moisture to the atmosphere. In addition to providing an energetic link between the surface and the atmosphere, the moisture flux links the water and the energy cycles (e.g., Jackson et al., 2009; Kubota and Tsutomu, 2008; Fajber et al., 2023). Despite their importance, evaporative (equivalently latent heat) fluxes are difficult to measure, and they are





one of the both largest and most uncertain terms in the surface energy balance (e.g., Liman et al., 2018; Clayson et al., 2019). An improved ability to quantify evaporative fluxes is therefore essential for observation based studies of the water and energy cycles, and the dynamics of weather systems and circulations that they fuel.

These fluxes can be reasonably well estimated directly, as the covariance of anomalies in moisture $q'$ and vertical air motions $w'$, i.e., as $\rho \ell_v \overline{w'q'}$, with $\rho$ the density and $\ell_v$ the vaporization enthalpy. Surface layer similarity provides a mean field theory

for the evaporative flux, which as encapsulated by the bulk aerodynamic formula, taking the form

$$\overline{w'q'} = C|U|\Delta q, \quad \text{where} \quad \Delta q \equiv q_s - q_a \tag{1}$$

so that the evaporative flux can be directly related to the specific humidity deficit of the air as compared to the surface, $\Delta q$, and the wind speed, with $C$ being an exchange coefficient. The value of the latter can depend on the surface properties and stability in a complex way, but it is well characterized by decades of careful measurements calibrating theoretical expectations

(Fairall et al., 1996, 2003; Edson et al., 2013). The surface moisture fluxes can therefore be reasonably well determined given knowledge of the specific humidity of the near-surface air, $q_a$, of the surface, $q_s$, and of the near-surface winds $|U|$.

Over water, where near-surface measurements are particularly sparse, the surface specific humidity can be estimated as the saturation specific humidity at the surface temperature and pressure, $q_*(T_s, P_s)$. Satellite remote sensing can provide reasonable estimates of $T_s$, which given $P_s$ determines $q_*$ and hence $q_s$. Likewise a variety of measurements provide increasingly

accurate estimates of surface wind speeds. The main limitation in estimating evaporative fluxes over the ocean is therefore the measurement of the near-surface specific humidity of the air, a quantity for which there is no real proxy. As a result satellite based climatologies of evaporative fluxes over the ocean depend on $q_a$ correlating with other quantities that can be remotely sensed, so that it (or the evaporative flux as whole) can be inferred statistically. Various approaches to this problem are detailed in Gentemann et al. (2020). These include retrievals from passive microwave measurements, such as the Hamburg Ocean

Atmosphere Parameters and Fluxes from Satellite (HOAPS4) (Liman et al., 2018; Andersson et al., 2010) and SeaFlux CDR (Clayson and Brown, 2016), as well as approaches that combine reanalysis and passive microwave data, such as IFREMER4 (Bentamy et al., 2013, 2017a) and J-OFURO3 (Tomita et al., 2019). Liman et al. (2018), for instance, compared HOAPS climatology with *in situ* buoy and ship measurements and found retrieval uncertainties in latent heat flux of $15\,\mathrm{W\,m^{-2}}$, with a global-mean error of 25 Wm$^{-2}$. Errors were found to be particularly large over the subtropical oceans, where evaporative fluxes

are large in magnitude, with an average of 37 Wm$^{-2}$ in random instantaneous retrieval (Liman et al., 2018) errors.

A number of studies have confirmed that the most uncertain term in Eq. 1 is $q_a$ (e.g., Bourras, 2006; Tomita and Kubota, 2006; Jackson et al., 2009; Bentamy et al., 2017b; Roberts et al., 2019; Robertson et al., 2020). Liman et al. (2018) estimated that contributions from $q_a$ contribute approximately 60% to overall uncertainty in the evaporative flux, whereas uncertainties from the wind speed contribute about 25%. And while uncertainties in $q_a$ are large over both land and ocean (e.g., Clayson

et al., 2019), our focus is on specific humidity measured over subtropical oceans, both because of large uncertainty in $q_a$ in this region (e.g., Jackson et al., 2009), and because over a sparsely instrumented land surface, estimating the surface specific humidity poses additional challenges.



To better estimate $q_a$, and hence evaporative fluxes over the ocean, we propose to use cloud base height as a proxy for the near-surface relative humidity of the air, $RH_a$. Over great expanses of the World Ocean, air temperatures are cooler than the underlying sea surface, as a positive sensible heat flux is required to balance radiative cooling within the boundary layer, and air typically flows from cooler toward warmer waters. This temperature difference gives rise to a well-mixed boundary layer topped by clouds. The height at which clouds begin to form — the cloud base — is a reliable indicator of the near-surface relative humidity. This fact has long been used in weather forecasting and flight meteorology and is demonstrated for conditions over the ocean in Sec. 3 using coincident lidar and sounding data from the EUREC⁴A field campaign (Stevens et al., 2021). In Sec. 4, we use airborne lidar measurements to demonstrate that cloud base height can also be reliably measured from a downward-looking lidar. This association anticipates the use of space based lidar remote sensing for this purpose. To use the bulk aerodynamic formula to estimate the evaporative flux requires an ability to infer $q_a$ from $RH_a$, and to measure $q_s$,. In Sec. 5 we show that this method is possible given knowledge of the surface temperature. We further test the proposed methodology using two case studies with airborne lidar data collected during EUREC⁴A. We demonstrate our proposed method's applicability to fair weather cloud regimes over the ocean, and hence its potential for direct use with satellite-based lidar, or as a way to augment already existing, but more empirical approaches. An overview of the data used in the aforementioned analysis is presented in Sec. 2. Discussion and conclusions are presented in Sec. 6.

## 2  Notation, data, and methods

Throughout we use s to denote surface quantities. The surface temperature of the ocean, $T_s$, is the skin temperature. The surface specific humidity, $q_s$ is equated with the saturation specific humidity, $q_*$ at $T_s$ and $P_s$. Near-surface atmospheric quantities are denoted by subscript a, to denote their value at a height $z_a$. We adopt the value of $z_a = 40\,\mathrm{m}$ above the surface, as this corresponds to the lowest reliable sonde measurements and is near the height of the air temperature measurements $(28.3\,\mathrm{m})$ made from the R/V Meteor and used in this study.

Air pressure is denoted by $P$ and vapor pressure by $e$, with $e_*$ denoting the saturation value for a plane of pure water. Hence, because the surface is water (albeit wavy and not pure), $e_s \approx e_*(T_s)$. To make it easier to manipulate in equations, for which abbreviations make poor symbols, we use the symbol $W$ to denote relative humidity. The symbol $\gamma_\eta$ is used to denote the vertical gradient of quantity $\eta$, so that $\gamma_W$ becomes shorthand for $\mathrm{d}W/\mathrm{d}z$.

We employ coincident sounding and lidar data from a variety of ground-based and airborne observing platforms during the EUREC⁴A field campaign (ElUcidating the RolE of Clouds, Circulation Coupling in Climate), which took place in January and February 2020 in the trade-wind zone east of Barbados (Stevens et al., 2021). During EUREC⁴A the German High Altitude and Long Range Research Aircraft (HALO) launched 810 dropsondes between January 22, 2020 and February 15, 2020 (George et al., 2021) (see the EUREC⁴A data paper for HALO by Konow et al. (2021)). These dropsondes yield vertical profiles of pressure, temperature, and relative humidity with a manufacturer-stated accuracy of 0.4 hPa, 0.1°C, and 2%, respectively (Vaisala, 2020). One unique aspect of EUREC⁴A is the sampling strategy that provides aggregated, statistical estimates of a larger-scale signal, compared to individual point-wise measurements. During EUREC⁴A, dropsonde measurements were





distributed along a fixed flight pattern, the 'EUREC⁴A circle' — a circular flight pattern with an approximately 220-kilometer diameter, centered at 13.3°N, 57.7°W, at 9.5 km altitude. Following Bony et al. (2017); Stevens et al. (2021), one *circle-mean* refers to the mean of typically 12 dropsondes launched over one hour along the EUREC⁴A circle (due to operator and instrument errors, on some circles fewer sondes were launched, but never fewer than seven). A *circling-mean* is defined as the

mean of three consecutive circle-means (or in two cases, two circle-means), corresponding to 30–36 consecutive soundings aggregated over 210 minutes.

Also from HALO, amospheric backscatter and water vapor DIfferential Absorption Lidar (DIAL) profiles were recorded using the airborne demonstrator for the WAter vapor Lidar Experiment in Space (WALES) (Wirth et al., 2009) on-board HALO. Due to its high horizontal and vertical resolution, airborne lidar data is amenable to studying small scale clouds, such

as in trade cumulus regions. Here we evaluate the lidar data at the highest possible resoltuion, e.g., the backscatter ratio and aerosol depolarization data are analyzed at 0.2 second time resolution and 7.5 m vertical resolution. Only data at a wavelength of 532 nm are used. The backscatter profiles are extinction corrected using the High Spectral Resolution Lidar (HSRL) method (Esselborn et al., 2008). All data are regridded to a constant altitude scale over the EGM96 geoid.

During the same period, radiosondes were launched as part of the campaign from the Barbados Cloud Observatory (BCO,

Stevens et al., 2016) and the research vessel, R/V Meteor Stephan et al. (2020), from January 16 to March 1, 2020. Ceilometer measurements were made from January 18, 2020 to February 14, 2020 from the same platforms, the BCO and R/V Meteor. At the BCO the ceilometer is an OTT CHM 15k pulsed laser cloud height detector at 1064 nm used to detect cloud base height and lifting condensation level; at the the R/V Meteor, the ceilometer is a Jenoptik system measuring vertical profiles of attenuated backscatter at 1064 nm to infer cloud base as a function of altitude and aerosol (Stevens et al., 2021).

**3  Cloud base height as a proxy for $RH_a$**

To evaluate the relationship between cloud base height, $h$, and $W_a$, we compare ceilometer estimates of $h$ with profiles of relative humidity. First, however, it is useful to develop theoretical expectations of how the the relative humidity should change with height. For a well-mixed layer and unsaturated layer the specific humidity is constant, and the temperature varies adiabatically. From the definition of relative humidity,

$$W = \frac{e_v}{e_*(T)} = q\frac{R_v}{R}\frac{P}{e_*(T)}, \tag{2}$$

where $R \equiv (1-q)R_d + qR_v$ is the gas constant of moist air.

For the adiabatic ascent of a parcel, with pressure changing hydrostatically following that of an environment at the same temperature, and with the change in $e_*$ given by the Clausius-Clapeyron equation, the vertical gradient of $W$, becomes:

$$\gamma_W \equiv \frac{dW}{dz} = \frac{g}{T}\left(\frac{\ell_v}{c_p R_v T} - \frac{1}{R}\right)W. \tag{3}$$

Eq.(3) implies that $\gamma_W$ increases with height, as both $T$ decreases and $W$ increases adiabatically. Averaging across a well mixed boundary layer with a depth of $600\,\mathrm{m}$, $q_v = 15\,\mathrm{g\,km^{-1}}$ and $T$ varying dry-adiabatically about a mid-layer ($300\,\mathrm{m}$) mean value





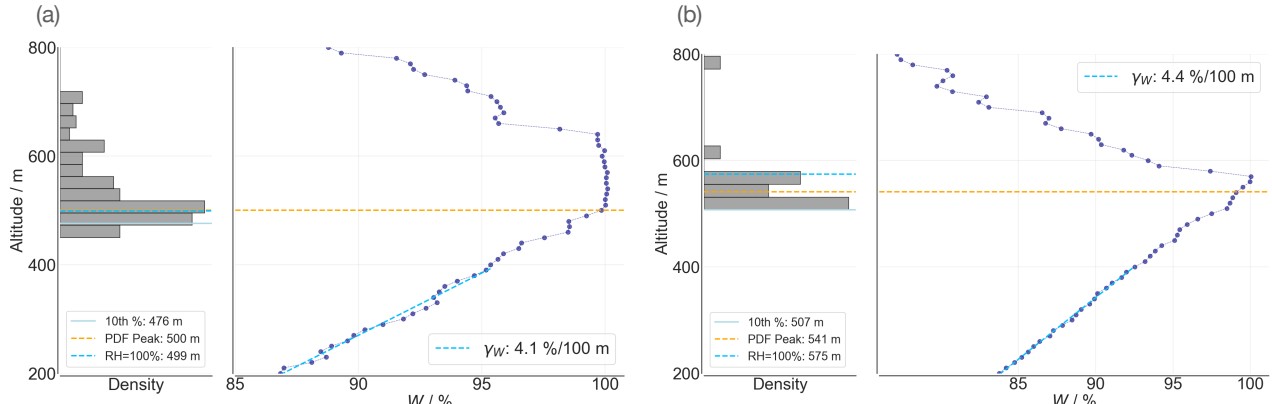

**Figure 1.** First ceilometer cloud returns for 30 minutes before and after the radiosonde launch time (grey histograms) and the relative humidity profiles measured by the radiosondes (blue profiles, data every $10\,\mathrm{m}$). Also shown on the ceilometer cloud base height histograms are the 10th percentile (solid light blue line), the first major peak of the distribution (orange dashed line), and the height at which relative humidity would reach 100% when extrapolating from a linear regression fit to the observed profile from 200-400 m (turquoise dashed line), as described in the text. Panel (a) is around the radiosonde launched on Jan. 29, 2020 at 06:44 UTC (2:44 am local Barbados time) and panel (b) is Feb. 1 2020 at 00:25 UTC (8:25 pm local Barbados time). These are two cases where radiosonde relative humidity reached 100% below $1000\,\mathrm{m}$.

of $296.41\,\mathrm{K}$ – values derived from the Meteor and BCO radiosondes – yields a boundary layer mean value of $\gamma_w = 4.0\,\%\,\mathrm{hm}^{-1}$ (referring to percent per hectometer, or $100\,\mathrm{m}$).

In reality the subcloud layer is not perfectly well mixed. Rising air parcels mix with descending air parcels that have incor-
porated air entrained from above the cloud layer. This causes the specific humidity of rising parcels to decrease slightly with altitude and their temperature to decrease with altitude less rapidly than for adiabatic ascent. To account for these processes we return to the definition of the relative humidity to note that

$$\mathrm{d}W = \left(\frac{\partial W}{\partial q}\right)\mathrm{d}q + \left(\frac{\partial W}{\partial P}\right)\mathrm{d}P + \left(\frac{\partial W}{\partial T}\right)\mathrm{d}T \tag{4}$$

with

$$\frac{\partial}{\partial q}W = \left(\frac{1}{q} - \frac{R_\mathrm{v} - R_\mathrm{d}}{R}\right)W \tag{5}$$

the relative humidity lapse rate becomes

$$\gamma_W = \left[\left(\frac{1}{q} - \frac{R_\mathrm{v} - R_\mathrm{d}}{R}\right)\gamma_q - \frac{g}{RT} - \frac{\ell_\mathrm{v}}{R_\mathrm{v}T^2}\gamma_T\right]W. \tag{6}$$

As a check we note that Eq. (6) reduces to Eq. (3) in the adiabatic limit, where $\gamma_q = 0$ and $\gamma_T = -g/c_p$, as it should. For the trade-wind layer as observed during EUREC⁴A, the data of Albright et al. (2022) suggest that $\gamma_q \approx 1\,\mathrm{g\,kg^{-1}\,km^{-1}}$, and



$\gamma_z \approx 9.4\,\mathrm{K\,km^{-1}}$ (Albright et al., 2022). Using these values in Eq. (6) and again taking $T = 293\,\mathrm{K}$ and $q = 15\,\mathrm{g\,kg^{-1}}$ yields $\frac{1}{W}\gamma_W \approx 4.0\,\%\,\mathrm{hm^{-1}}$, which is about 16% less than its adiabatic value, with most (85%) of the difference being explained by the reduction of $q$ with height.

For the data we use $W$ from radiosondes and cloud base height distributions are derived from ceilometer measurements during 60 minute windows centered around the radiosonde launch time (plus or minus 30 minutes from the launch time). From
the resulting distribution of ceilometer cloud detections, we associate $h$ with the first and major peak of the distribution, similar to the method employed in Albright et al. (2022) and Vogel et al. (2022). There is a strong correlation between $h$ and the 10[th] percentile (r=0.94) or other low quantiles. Associating cloud base with the main peak of the distribution (or low quantiles) accounts for the expectation that ceilometer based cloud detections are skewed to more elevated values (Nuijens et al., 2014). This finding reflects the tendency of clouds to dissipate from their base upwards, leaving cloud remnants to evaporate above
cloud base. Similarly a local maximum in the wind speed near cloud base results in cloud-base scudding ahead of more elevated regions of the cloud mass, which would also lead to a longer tail of more elevated ceilometer cloud returns. Rain, on the other hand, is infrequent and not readily identified in the ceilometer signal, leading less often to the situation whereby ceilometer estimates of $h$ are low-biased (Nuijens et al., 2014). Fig. 1 illustrates this method for two example 60 minute periods for radiosondes where relative humidity reached 100% below 1000 m. First ceilometer cloud detections are plotted as a
histogram for the two cases, along with associated radiosonde profiles of relative humidity. For the ceilometer cloud base height distribution, horizontal lines illustrate different choices of $h$: the 10[th] percentile, first distribution peak, and the extrapolation to the altitude where $W$ reaches 100% based on linear fits to $W$ in the layer between 200-400 m.

Using the first distribution peak of ceilometer values to estimate $h$, we calculate $h$ and $W_\mathrm{a}$ for 118 radiosondes at the BCO and 171 radiosondes at the R/V Meteor. Fig. 2 shows the close association between these two quantities. Also shown for
comparison are subcloud layer height estimates from dropsonde measurements and virtual potential temperature, $\theta_\mathrm{v}$, vertical profiles (averaged at the $\sim$220 km diameter circle spatial scale, over three hours, following Vogel et al. (2022) and Albright et al. (2022), see Sec. 2). Theoretical estimates from Eq. 6 are also plotted, both the adiabatic case and the case where $\gamma_q$ and $\gamma_T$ are allowed to depart from their adiabatic profile. The agreement between the lines and the points in Fig. 2 show the expected consistency.

## 4  Cloud base detection from airborne lidar

Having established the relationship between $h$ and $W_\mathrm{a}$ based on both theory and surface measurements, we now turn to estimating the relationship using data from WALES airborne data. For each profile, the lowest altitude where the lidar signal is above background aerosol values is selected using a backscatter ratio of 20. We only consider cases where the sea surface is still visible, which ensures the accuracy of the extinction correction by the HSRL method (see Sec. 2). This limits detections to
optically thin clouds or the corner regions of clouds, as most, deeper clouds are opaque to the downward-staring lidar, and has the advantage of being less susceptible to rain detections. For edge cases this implies that the cloud base near the edges of the clouds can be extrapolated to the center of the cloud, which is opaque to the lidar (e.g., assuming that the clouds are mostly flat



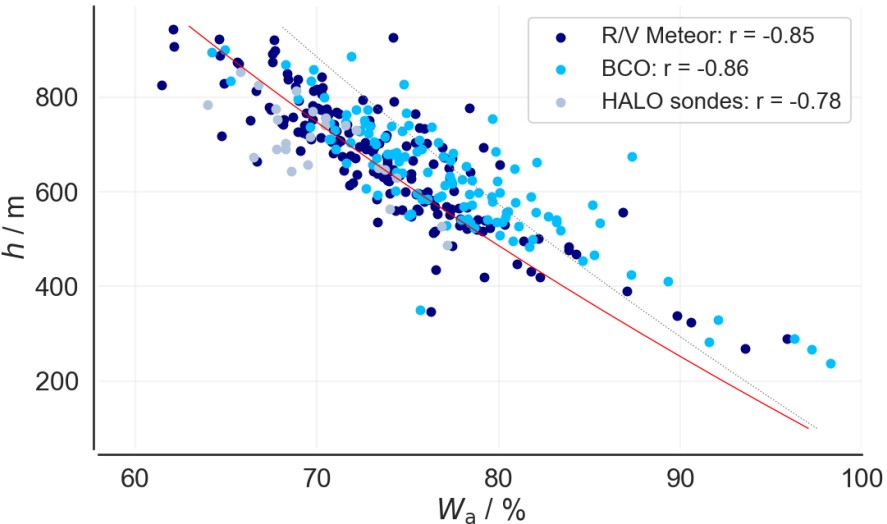

**Figure 2.** $W_a$ from radiosondes and ceilometer-based estimates of $h$ (using the first, major peak of the distribution) from R/V Meteor (n=171, dark blue) and BCO (n=118, medium blue) measurements. Also shown are area averaged estimates of $h$ from HALO dropsondes as calculated by Vogel et al. (2022) and Albright et al. (2022). Lines are theoretical relationships, as described in the text (red: adiabatic lapse rate of -9.8 K/km; light grey dashed line: temperature lapse rate of -8.5 K/km).

at the bottom). For multi-layered cloud systems, the base of the upper layer is sometimes identified instead of the lowest cloud base. These cases could be flagged and removed based on a cloud base height histogram controlled filter which uses the fact

that for multilayer systems a second or third mode appears. The two cases presented here do not show such upper layers and no additional filtering was applied. As discussed above in the case of surface-based measurements, 3D effects can also bias cloud bases high when the cloud is vertically skewed by wind shear, a situation that Nuijens et al. (2014) showed was not uncommon for the winter trades near Barbados. In this case the lidar will at times only intersects the top region. We minimize these effects by applying a gliding minimum filter with a width of $3\,\mathrm{km}$.

Fig. 3 presents results for two days of the campaign: January 28, 2020 (57 WALES lidar–dropsonde pairings) and February 2, 2020 (32 pairings), which are selected because they sampled a representative range of different conditions. January 28 was characterized by small cumulus clouds, often referred to as 'sugar' clouds (e.g., Stevens et al., 2020; Bony et al., 2020), while February 2 was characterized by deeper clouds with more stratiform layers near cloud top, often referred to as 'flower' clouds (e.g., Stevens et al., 2020) and the presence of a strong Saharan dust layer reaching up to $2.5\,\mathrm{km}$ . Overall there is a

good correspondence between $h$ as measured by WALES and the dropsonde $W_a$, all the more so given that the $W_a$ is a point estimate not necessarily centered on the lidar estimates.

Further analysis of the exceptions helps give confidence in the rule, i.e., the purported $W_a(h)$ relation. For the February 2, 2020 case, which had more stratiform clouds, three outliers are identified. To investigate these outliers, we examine the



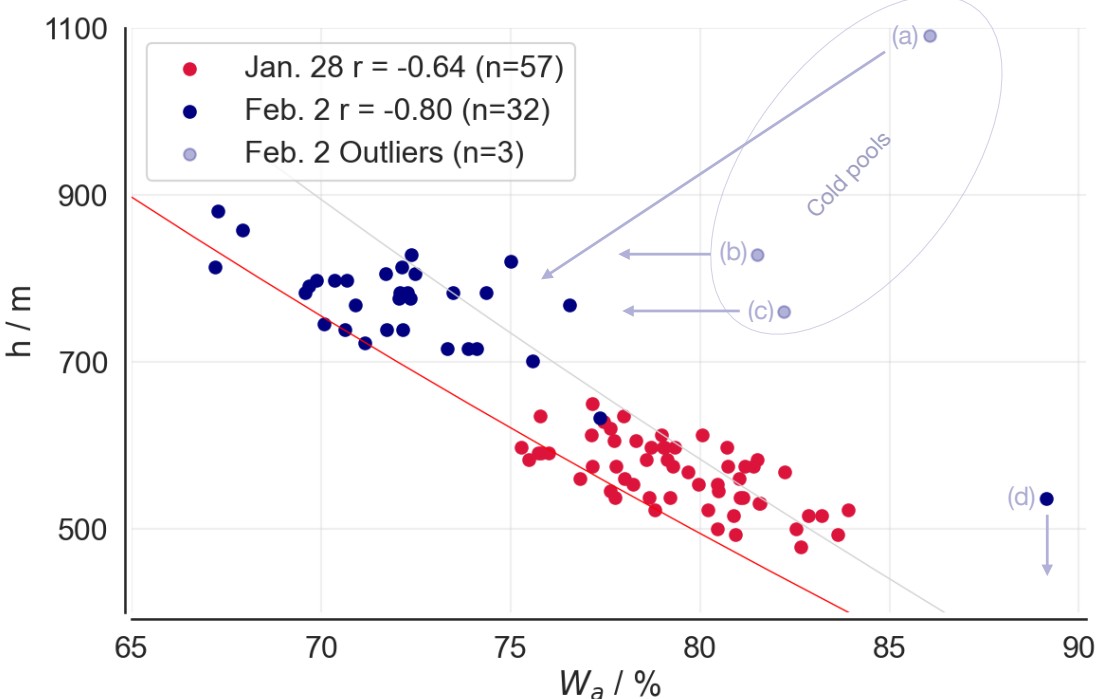

**Figure 3.** $h$ from WALES for two days, January 28, 2020 (red) and February 2, 2020 (blue), and $W_a$ from the nearest dropsonde launch. Outliers are labeled, with the suspected source of bias as suggested in the text. As in Fig. 2, lines are theoretical relationships (red: adiabatic lapse rate of -9.8 K/km; light grey line: temperature lapse rate of -8.5 K/km).

backscatter data used to estimate the cloud base height, vertical profiles of relative humidity for the closest dropsonde and the
dropsondes immediately prior and after, and visible satellite images of the cloud formations (Fig. 4). A visual inspection of
the scenes around the anomalous points suggest that the outliers are associated with cold pools that increase relative humidity
(increasing specific humidity and decreasing temperature) (Touzé-Peiffer et al., 2022), and/or cloud fragments associated with
dissipating or stratiform cloud elements. The outlier labeled (a) appears to be a cloud fragment from the stratiform cloud layer,
as seen in the backscatter ratios and, to some extent, in the visible satellite image; it is associated with higher relative humidity
than the dropsonde sounding immediately before and after. If instead estimating a cloud base height around 750 m and relative
humidity between 70–75% representative of the larger environment, the point would lie on the canonical curve, as illustrated
schematically in Fig. 3. Cases (b) and (c) correctly identify the cloud base height, but the cold pool soundings have higher-
than-expected relative humidity as seen in the dropsonde profiles. This high value of $W_a$ result in a $W_a(h)$ relationship that
differs from the expected linear relationship, but these values would not be expected to be associated with significant errors in
the inferred relative humidity because the cloud base height estimate is not biased. Having removed these outliers on Feb. 2,
the WALES lidar method has similar skill to ground-based lidar estimates as shown in Fig. 2.



**Figure 4.** Four anomalous $W_a$, $h$ pairings from Fig. 3. Shown for the four outlier points are: (left column) the backscatter ratio including the selected time as vertical line (red), the inferred cloud base height (blue horizontal line), and for the top panel, an approximate cloud base height value expected from the linear relationship (orange horizontal line); (center column) relative humidity profiles for the nearest sounding in blue, as well as the soundings immediately before and after (black, solid and dashed, respectively); (right column) visible satellite images (GOES-16 ABI) showing the cloud structure from above and the location of the dropsonde at its launch time, accessed via https://observations.ipsl.fr/aeris/eurec4a-data/PRODUCTS/GOES-E_movies/VIS_IR_combined/v1.0.0/.

 

In addition, point (d) in the figure appears to be associated with cloud forming on a cold pool boundary, which was uncharacteristic of the broader cloud environment (Fig. 4d). Here again it appears that the value does not violate the $W_a(h)$ relationship being used to infer $W_a$ from $h$, but rather the correspondence between the humidity estimate and the lidar selection of the low-

est cloud base. Recalculating the correlation with a cloud base height value of 300 m, the association increases (to r = -0.86). This example suggests some ambiguity in the estimate of $W_a$ based on the spread of cloud base estimates below the peak value of the distribution, something that would have to be fine tuned in an operational retrieval.

## 5   Estimating the air-sea vapor deficit

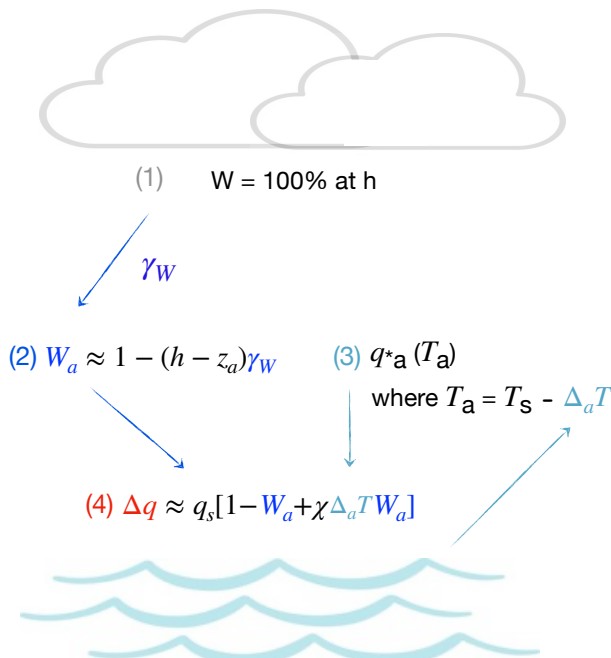

**Figure 5.** Schematic overview of the methodology highlighting the three main error sources: (1) estimating cloud base height, $h$, where W=100% is assumed and then extrapolated to $z_a$ using (2) a fixed relative humidity lapse rate $\gamma_W$, which is then converted to $q_a$ as described in the text with a fixed $\Delta_a T = -1.3\,\mathrm{K}$.

### 5.1   Theory

Having demonstrated the theoretical and empirical relationship between cloud base height and near-surface relative humidity from both ground-based and airborne measurements, we now focus on estimating the specific humidity deficit of the near-surface air, $\Delta q$.



Given $h$, this estimation will require a specification of $\gamma_W$ and the sea-air temperature difference $\Delta T$. The method can be demonstrated as follows: From the definition of $\gamma_W$, $W_a$ can be expressed as

$$W_a \approx 1 - (h - z_a)\gamma_W. \tag{7}$$

It follows from Eq. (2) that the surface humidity deficit can be calculated as

$$\Delta q = q_s \left\{ 1 - W_a \left[ \frac{e_*(T_a)}{e_*(T_s)} \right] \frac{P_s - [1 - R_d/R_v]e_*(T_s)}{P_a - [1 - R_d/R_v]W_a e_*(T_a)} \right\}. \tag{8}$$

With the help of the Clausius-Clapeyron Equation, we can linearize $e_*(T)$ about $T_s$, such that

$$e_*(T_a) = e_*(T_s)[1 - \chi\Delta T] \quad \text{where} \quad \chi \equiv \frac{\ell_v}{R_v T_s^2} \approx 1/16 \, \text{K}^{-1}. \tag{9}$$

With this approximation and by neglecting small changes from the corrections due to differences in the air pressure (equivalently setting the rightmost fraction in Eq. (8) to 1), we arrive at the comparatively simple expression for the specific humidity deficit,

$$\Delta q \approx q_s [1 - W_a + \chi\Delta T W_a]. \tag{10}$$

Substituting for $W_a$ from Eq. (7) thus shows how $h$, $\gamma_W$ and $\Delta_a T$ enter into the estimate of $\Delta q$. As such, Eq. (10) also lends itself well to analytic estimates of the error. For a relative error $\varepsilon_h$ in estimating $h\gamma_W$, and a relative error $\varepsilon_T$ in estimating $\Delta T$, the relative error in the specific humidity deficit of the near-surface air becomes

$$\varepsilon_q \approx W_a \varepsilon_h + \frac{\chi\Delta T W_a}{1 - (1 - \chi\Delta_a T)W_a}[\varepsilon_h + \varepsilon_T]. \tag{11}$$

By using Eq. 8 to directly calculate $\Delta q$ we calculate that a $1\%$ error in $h\gamma_W$, $\Delta_a T$ and $P_s - P_a$, with $W_a = 0.72$ gives a $\varepsilon_q$ of $0.84\%$, $0.13\%$, and $0.01\%$. Using Eq. (11) yields values of $0.85\%$, $0.1\%$, and $0\%$ respectively.

This analysis confirms what what Eq. (11) is trying to tell us, which is that the main limitation to accurate estimates of $\Delta q$ is the accuracy of the estimate of the product between $h$ and $\gamma_W$. From the point of view of the energy budget, which also depends on the sensible heat flux, an error in $\Delta_a T$ will also lead to errors of a similar sign in estiates of that quantity. The net energy flux error will then depend on th e Bowen ratio, $B$, and increase with decreasing $B$ and increasing $\varepsilon_T$ as $(1 + 0.13/B)\varepsilon_T$. Over the ocean $B \approx 0.05$–$0.30$, this factor ($3.6$ down to $1.4$) is however still substantially smaller than it would need to be for the error from $\varepsilon_T$ to become commensurate with that associated with a similar value of $\varepsilon_h$.

## 5.2 Estimates of $\gamma_W$ and $\Delta_a T$

Fig. 6 shows the relative humidity lapse rate, $\gamma_W$ calculated from R/V Meteor radiosondes from January 16 to March 1, 2020. We first segment into cloudy soundings (defined as soundings where $\gamma_W$ reaches 100% below 1000 m) and non-cloudy soundings and then calculate $\gamma_W$ between 200–400 m. For cloudy soundings, occurring about 10% of the time, the mean is $3.9\,\%\,\text{hm}^{-1}$ (median $3.8\,\%\,\text{hm}^{-1}$, $2.4\,\%\,\text{hm}^{-1}$ to $4.7\,\%\,\text{hm}^{-1}$ 5-95% quantiles), whereas for non-cloudy soundings, the





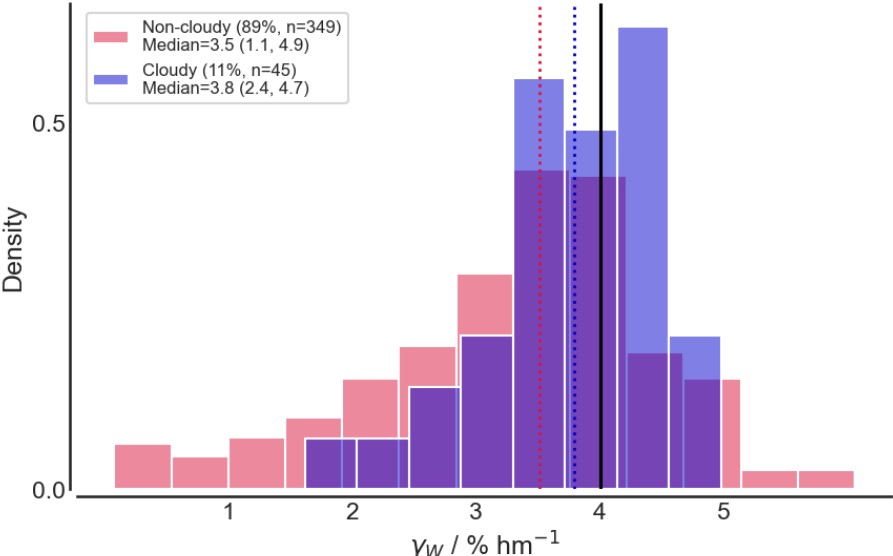

**Figure 6.** Histograms of $\gamma_W$ calculated from R/V Meteor soundings as a linear regression between $200\,\mathrm{m}$ to $400\,\mathrm{m}$: cloudy soundings (blue), defined as soundings where $W \geq 100\%$ below $1000$ m), and non-cloudy soundings (red) otherwise. Vertical dashed lines are the median of each distribution, and the solid black line is the value of $4\,\%\,\mathrm{hm}^{-1}$ selected as a representative relative humidity lapse rate.

distribution is broader with a median of $3.5\,\%\,\mathrm{hm}^{-1}$ ($1.1\,\%\,\mathrm{hm}^{-1}$ to $4.9\,\%\,\mathrm{hm}^{-1}$, 5-95% quantiles). This method is also illustrated in Fig. 1. These data show that the adiabatic lapse rate of $\gamma_W$ provides a good upper bound. Based on these cases, and the calculations above, in the following we assume $\gamma_W = 4\,\%\,\mathrm{hm}^{-1}$, a value close to the distribution peak of the cloudy soundings and to the expected theoretical value (Sec. 2).

For $z_a = 40\,\mathrm{m}$, $\delta_a P$ is negligible compared to $P$. Given $W_a$ we can thus calculate $q_s - q_a$ given $h\gamma_W$ and $\Delta_a T$. We showed that $\gamma_W$ is expected to be relatively constant, similar arguments lead us to expect $\delta_a T$ to also be relatively fixed. This is because it too is controlled by the dynamics of the convective boundary layer, which is constrained by the well mixedness and hence near adiabaticity of the temperature profile, and the radiative cooling of the subcloud layer.

Fig. 7a shows the time evolution of sea surface temperatures measured by a thermosalinograph from the port sensor at a depth
of $5\,\mathrm{m}$ and near-surface air temperatures at the mast height ($28.3\,\mathrm{m}$) measured by the R/V Meteor. The former is expected to be biased warm as compared to the ocean skin temperature due to the cool skin effect. Typical values for this effect are between $0.1\,\mathrm{K}$ to $0.3\,\mathrm{K}$. As expected, almost all (97%) of the 47,512 measurements have a warmer ocean than near-surface temperature, such that the atmosphere is unstable and conducive to convection and cloud formation. The median and mean differences between ocean and near-surface temperatures are 1 and 1.1 K, respectively. The temperature that is, however, relevant for
surface fluxes and the near-surface temperature offset is the sea surface *skin* temperature, which we take to be $0.3\,\mathrm{K}$ colder than the sea water temperatures measured at a depth of a few meters. We therefore test the assumption of using a fixed offset of $-1.3\,\mathrm{K}$ from the measured, time-varying sea surface temperatures associated with the radiosonde launch times.




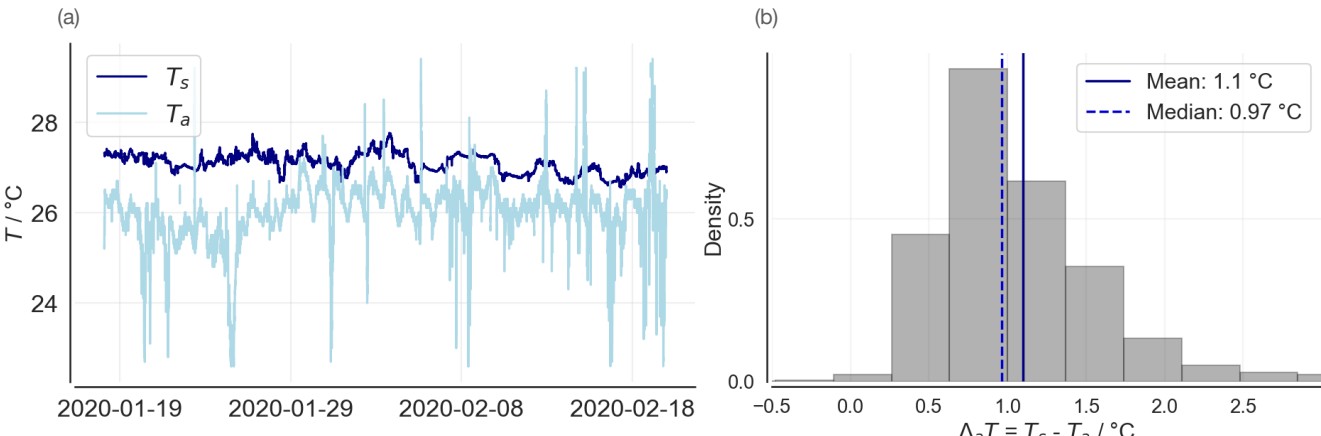

**Figure 7.** (a) Time series of sea surface temperatures ($T_s$), measured 5 m below the ocean surface, and near-surface air temperatures ($T_a$) measured at 28.3 m at the backboard of the R/V Meteor. (b) The distribution of differences, $\Delta_a T$, between the sea surface and near-surface air temperature, with vertical lines for the mean and median values.

## 5.3 Near-surface specific humidity estimated from ground-based and airborne lidar cloud base estimates

Fig. 8 (top row) presents the time series of predicted near-surface specific humidity, $q_a$, based on 171 co-located ceilome-
ter–radiosonde pairs and observed radiosondes launched from the R/V Meteor. On average the cloud base height method overestimates $q_a$ by $0.33\,\mathrm{g\,kg^{-1}}$ (5th–95th percentile range: $-0.74\,\mathrm{g\,kg^{-1}}$ to $1.8\,\mathrm{g\,kg^{-1}}$), with a median absolute error of $0.47\,\mathrm{g\,kg^{-1}}$. The temporal variability is well captured ($r = 0.76$). The largest positive biases occur early in the campaign when cloud bases were low – the error in $q_a$ correlates with cloud-base height ($r = 0.54$ overall, rising to $r = 0.76$ for bases below 600 m), suggesting that shallow, poorly mixed layers (e.g., beneath decaying cold pools; see Sec. 4) are less amenable
to this approximation. No systematic diurnal bias is evident, although the small sample size limits a definitive assessment of hour-of-day effects.

Fig. 8 (bottom row) shows the comparison between WALES airborne-lidar–derived $q_a$ estimates and coincident dropsonde measurements also from HALO. Here the cloud-base method again reproduces the variability reasonably well ($r = 0.61$), with a small mean bias of $-0.06\,\mathrm{g\,kg^{-1}}$ and a median absolute error of $0.26\,\mathrm{g\,kg^{-1}}$ on January 28, 2020. On February 2, 2020, the
correlation is slightly lower ($r = 0.57$), with a mean bias of $-0.03\,\mathrm{g\,kg^{-1}}$ and a median absolute error of $0.27\,\mathrm{g\,kg^{-1}}$.

## 6 Discussion and conclusions

We have demonstrated that downward-staring lidar estimates of cloud base height can be used to estimate near-surface relative humidity. Combined with estimates of wind speed, surface temperature, and surface-air temperature, knowledge of cloud base height thus provides a basis for estimating the surface latent heat flux. Over the ocean, where such measurements are most





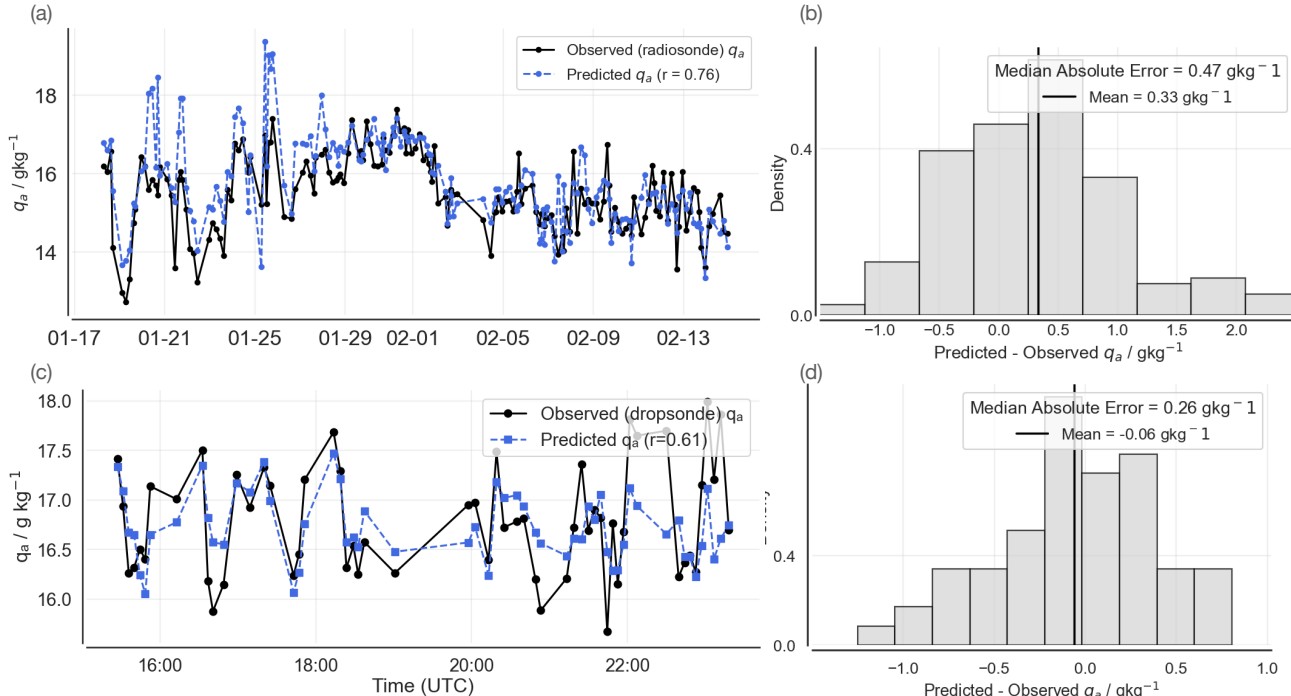

**Figure 8.** Comparison of observed and predicted near-surface specific humidity estimates and their error distributions. (a) Time series of observed $q_a$ from radiosondes launched from the R/V Meteor (solid black) versus the cloud-base-height-derived prediction from R/V Meteor ceilometer data (dashed blue; $r = 0.76$). Time is month and day. (b) Histogram of predicted minus observed $q_a$ for the R/V Meteor data with a median absolute error of $0.47\,\mathrm{g\,kg^{-1}}$ and a mean bias of $0.33\,\mathrm{g\,kg^{-1}}$ (vertical line). (c) Time series of observed $q_a$ from HALO dropsondes (solid black) versus cloud-base-height-derived predictions from WALES airborne lidar estimates on January 28, 2020 (dashed blue, $r = 0.64$). Time is hour and minute, UTC. (d) Histogram of predicted minus observed $q_a$ for the HALO comparison, with a median absolute error of $0.26\,\mathrm{g\,kg^{-1}}$ and a mean bias of $-0.06\,\mathrm{g\,kg^{-1}}$ (vertical line).

lacking, and most important for the surface energy balance, scatterometer measurements provide estimates of the surface wind speed, and a variety of instruments having long provided satellite-derived estimates of surface temperature. Over most of the global ocean, the air-sea temperature difference varies over a small range, so that it could be sufficiently well estimated using statistical methods. Estimates of near-surface relative humidity from estimates of cloud base could thus provide the missing ingredient for physically-based remote sensing estimates of the surface latent heat flux, globally.

275       The method we propose for estimating near-surface humidity requires unbiased estimates of cloud base height, and the satisfaction of two further assumptions: (i) that the relative humidity lapse rate is near the value it would obtain in a well-mixed layer, and thus relatively constant; and (ii) that the near-surface air temperature is cooler than the surface, so that the layer above is convectively driven. Convective boundary layer clouds – which form in the radiatively cooled, cold advection-dominated boundary layers that prevail over tropical oceans – both underpin our near-surface humidity estimates and confirm that the





very conditions required for those estimates are in place. While this physical situation limits the application of the method to conditions where shallow convective clouds are present, they are ubiquitous, even in regions of deep convection. Our analysis shows that the method requires some calibration for estimating cloud base from a distribution of lidar echos, for estimating the relative humidity lapse rate, and for estimating the air-sea temperature difference. Their unbiased estimation will be required for using the proposed method to establish large-scale climatologies of near-surface relative humidity and associated latent

heat fluxes.

As an outlook we note that the method we propose would facilitate the development of a long-term, day and night, and physically-based near-surface humidity climatology if applied to global data of cloud base height. Techniques using multi-angle satellite imagery have been shown to retrieve cloud base height, albeit over longer timescales (Böhm et al., 2019). The most promising candidate for obtaining such data is measurements using a spaceborne lidar. Currently the newly-launched

EarthCARE satellite (Illingworth et al., 2015) provides HSR-Lidar data. It provides backscatter data with a horizontal resolution of about $280\,\mathrm{m}$ and a vertical resolution of $100\,\mathrm{m}$. While the horizontal resolution appears to be sufficient to apply our method, the limited vertical resolution would imply an error of the near-surface humidity of about $0.5\,\mathrm{g\,kg^{-1}}$ attributed to the vertical sampling error alone (based on Eq. (11) assuming typical values for $W_\mathrm{a}, h$, etc.). With sufficient sampling, and given the variability of cloud base height, it might be possible to obtain greater precision in estimates of the mean cloud base height than

what is implied by the single-snapshot vertical resolution. Laser ranging using more sophisticated methods, such as employed by the Global Ecosystem Dynamics Investigation lidar aboard the International Space Station, could provide better estimates of cloud base height, but presently does not provide products that allow this capability to be explored. Future satellites refining these technologies could be adapted to the requirements of our proposed method, and potentially could provide coincident estimates of near-surface wind speed and air-sea temperature difference. In this context, measurements of ocean surface texture

using synthetic aperture radar could also be explored as a way to estimate the difference between the surface temperature and that of the air just above it, which would better constrain the proposed estimates both directly, and indirectly due to the expected covariabilty of the relative humidity lapse rate and the air-sea temperature difference. At the very least our work suggests that cloud base height information from global lidar measurements might be usefully incorporated in reanalyses and other, more statistical, approaches.

*Author contributions.* Anna Lea Albright and Bjorn Stevens are co-first authors. The original idea was proposed by Bjorn Stevens and developed together with Anna Lea Albright. Anna Lea Albright wrote the original draft, performed most of the analysis, and drafted the figures. Martin Wirth prepared and analyzed the WALES data and contributed to the implementation. Bjorn Stevens and Martin Wirth both contributed to the writing of subsequent manuscript drafts.

*Competing interests.* The authors declare no competing interests.



*Acknowledgements.* Anna Lea Albright thanks the Max-Planck-Gesellschaft for travel and visitor support. She acknowledges using artificial intelligence to help edit code and text.



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
