# Peer review of "Estimating near-surface specific humidity over the ocean"

_EGUsphere, 2025_

## Author Comment (AC1)

Dear Editor and Reviewers,

We thank the reviewers for their constructive and thoughtful feedback. In this cover letter, we address three central themes raised and then respond to comments line by line.

We, moreover, conducted an independent review of our manuscript using ChatGPT. This review raised similar points to those raised in the reviewer comments. Most pointedly, ChatGPT also suggested a transfer to AMT, thus highlighting what is likely a general disposition of the manuscript, something we kept in mind in addressing the reviewer comments.

**1) Scope and applicability of the method**

The paper advances a physical idea: that, in the convective marine boundary layer, cloud base height contains first-order information about near-surface humidity. We appreciate that the original framing might not have made this limitation clear, as we took for granted that a convective boundary layer is the dominant condition over the ocean. We now make this condition more explicit by including a global 'buoyancy-favorability' map (Fig. 1), which shows the climatological prevalence of near-surface positive buoyancy of air over the ocean (a proxy for convective, well-mixed layers).

New Figure 1:

Caption: Frequency of days with positive surface buoyancy flux, \$B\_0\$, during 2020, computed from daily ERA5 reanalysis data of sensible and latent heat fluxes, as well as the 2\,m air temperature. The shading indicates the fraction of days for which the surface buoyancy flux is positive (upward). Positive values represent convectively unstable surface conditions. Values approach unity over most tropical and subtropical ocean regions, indicating nearly continuous convective instability. Over the global oceans, the area-weighted mean frequency is approximately 85\%, while in the tropical band (30\$^\circ\$\,S--30\$^\circ\$\,N) it reaches 99\%. White contours denote frequencies of 0.50, 0.75, 0.90, 0.95, and 0.99.

In this spirit, we also revised the title to: "Estimating near-surface specific humidity over convective oceanic regions from cloud base height observations".

**2) Uncertainty, sensitivity, and calibration**

This manuscript's goal is to introduce and test a physical retrieval concept and identify its dominant sensitivities. An uncertainty quantification will depend on the instrument characteristics and the implementation of the algorithm. Since we do not have the instrument, we cannot, at this stage, do the implementation. As such, we focus on what parameters need to be considered given an instrument and an implementation, and their relative importance.

To make this more apparent for the reader, we moved the schematic and discussion of parameter sensitivities earlier, and introduced a new Section 3, "Theory and parameter sensitivities for cloud base height as a proxy for near-surface humidity". Also relating to point 1) above, we added a new section 5, "Scope, caveats, and practical use," which clarifies when the approach is valid and where it fails (e.g., cold pools; multilayer or optically thick clouds; shallow, poorly mixed layers). Because additional case studies would not advance the core physical idea, we instead summarize the key uncertainties and the calibration needed for an operational product.

**3) Journal fit**

We respectfully maintain that the paper is centered on a physical retrieval concept and its mechanistic underpinnings, not on a new measurement technique or technology. It connects boundary layer thermodynamics to a practical path for constraining near-surface humidity and thereby the surface latent heat flux. This physical basis, and the time already invested in reviewing this manuscript for ACP, motivate our desire to bring this manuscript to a conclusion within ACP. We nonetheless appreciate the reviewers' perspective and would defer to the editor on this point.

We believe these revisions address the reviewers' core concerns while preserving the paper's central contribution: an empirically and theoretically supported physical link between cloud base height and near-surface humidity in convective marine boundary layers, alongside a roadmap for error quantification and turning the idea into a calibrated product.

We thank the reviewers and the editor for their time and insights.

Thank you and best regards,

Anna Lea Albright on behalf of all authors

**Reviewer comments:**

**Reviewer 1:**

This paper outlines a relatively simple method of retrieving near-surface humidity as a function of cloud-base height assuming an adiabatic and well-mixed layer below the clouds. The theorized relationship makes sense and a decent amount of data supports the authors' points. However, there are some fundamental issues here which need to be addressed before this paper can be published. Therefore, I am recommending major revisions as well as a transfer to a different journal.

Fundamentally, this paper assumes that the observed cloud base height can be used to infer the surface and near-surface relative humidity through simple adiabatic thermodynamics. This works if the boundary layer is well-mixed and the clouds are convective in nature as the clouds are coupled to the PBL. For the EUREC4A campaign location of Barbados, those conditions may very well dominate. However, this paper casts a much larger net than the tropics. The title and abstract make no restrictions on the time and place where this approach can be applied. Given that stratocumulus are by far the most common marine cloud type, and the formation of these clouds is largely independent of PBL mixing, the utility of the proposed method is much more limited than what the paper implies. It still may have utility (notwithstanding further critiques below) but the locations where its use is appropriate need to be identified from the very start. The authors partially mitigate this with a statement around Line 278, but even then they are focused on areas of convection and not the more prevalent stratiform clouds. In Line 259 they note that error increases in shallow, poorly mixed layers. That's much of the globe.

To address this reviewer comment about applicability globally, we have added a new Figure 1 that shows the ubiquity of a convective cloud-topped boundary layer over the world oceans, as described in point 1) in the above cover letter. Furthermore, we have added the following text starting line 67: "We demonstrate this ubiquity by analyzing daily ERA5 surface fluxes from the year 2020 to compute the climatological frequency of positive surface buoyancy flux, \$B\_0\$, representing the annual frequency of convectively unstable surface conditions. The resulting `buoyancy-favorability' map (Fig.~\ref{fig:buoyancy\_flux\_frequency}) shows that near-surface convective instability prevails over most tropical and subtropical oceans, with ocean-only mean frequencies

of 85\% globally and 99\% between 30\$^\circ\$\,S and 30\$^\circ\$\,N." Please see new Figure 1 in the above cover letter.

We have also revised the title to address this reviewer comment about specifying the conditions under which we would expect the method to hold, with the new title being "Estimating near-surface specific humidity over convective oceanic regions from cloud base height observations".

Lastly, we also added a new section 5, "Scope, caveats, and practical use" to outline the conditions under which the method would be expected to have less skill.

The thesis of the entire paper rests on a single sentence, found in line 63: "The height at which clouds begin to form - the cloud base - is a reliable indicator of the near-surface relative humidity." This is a rather foundational statement that the authors do not support with any references. The authors note that this is effectively used as a rule of thumb, but many such rules are unsupported by scientific evidence, and it is important to verify the validity of those rules when used in a scientific application. Thus, the authors are obligated to support the crux of their argument with evidence, but it is largely lacking here. The causality between cloud base height (for all clouds, mind you, since there is no effort to specifically segregate convective clouds from others here) must be identified and supported if the authors are going to make such a statement. I realize that Fig. 2 is an attempt to empirically show this relationship, but again this will only be true for convective clouds. Even at that, the relationships shown in Fig. 2 are more tenuous than one might hope: the BCO radiosondes in Fig. 2 exhibit an r2 of approximately 60%. That's quite a lot of variance that cannot be explained by a simple relationship between the two variables, and this will introduce a substantial amount of uncertainty in the final product. It's not clear with that much variability that cloud base is truly a "reliable indicator of near-surface relative humidity."

We thank the reviewer for this comment. We agree that the relationship between cloud base height and near-surface relative humidity is foundational to our approach and warrants clear justification. We have added additional text to the introduction (around line 63) to discuss the basis of this relationship – which we then test empirically in the remainder of the paper.

"Given this uncertainty, our goal is to develop a method to estimate q\$\_\ra\$ over convective oceanic regions. To this end, we exploit the physical connection between cloud-base height \(h\) and near-surface relative humidity \(\RH\_\ra\): in a convective,

well-mixed subcloud layer the cloud base forms near the lifting condensation level and thus the height at which it forms depends primarily on near-surface \((T\)) and \((q\)).

Our method takes advantage of the fact that a convective cloud-topped boundary layer is ubiquitous over the world ocean (Fig.~\ref{fig:buoyancy\_flux\_frequency}). We demonstrate this ubiquity by analyzing daily ERA5 surface fluxes from the year 2020 to compute the climatological frequency of positive surface buoyancy flux, \$B\_0\$, representing the annual frequency of convectively unstable surface conditions. The resulting `buoyancy-favorability' map (Fig.~\ref{fig:buoyancy\_flux\_frequency}) shows that near-surface convective instability prevails over most tropical and subtropical oceans, with ocean-only mean frequencies of 85\% globally and 99\% between 30\$^\circ\$\,S and 30\$^\circ\$\,N.

Building on this link, we test the idea that  $\(\RH_\ra\)$  (and hence  $\(q_\ra\)$ ) can be inferred from  $\(h\)$  and a small set of parameters, as a basis for a possible retrieval. To this end, we summarize notation and data (Sec.~\ref{sec:notation\_data}); we then derive the relationship between  $\(h\)$ ,  $\(\RH_\ra\)$ , and  $\(\Delta\ q \equiv\ q_\rs - q_\ra\)$  and quantify sources of uncertainty (Sec.~\ref{sec:theory})."

The mathematics of the paper rely on the claim that the surface air is saturated. This is a statement that left me scratching my head as I read it. Obviously on a bulk level this statement is false: there is no permanent fog layer across each body of water on Earth. But even on a molecular level, is this true? At the ocean skin, is the number of evaporating H2O molecules really always equally balanced by the number of condensing ones? I could be mistaken on this, but it is incumbent upon the authors to justify this statement. Because of this, it calls into question the rest of the analysis that flows from that assumption.

We thank the reviewer for this comment. We agree that, in a literal sense, the air immediately above the ocean surface is not permanently saturated —there is of course no permanent fog layer over the open ocean. However, in the context of bulk aerodynamic formulations of surface fluxes, the assumption that the surface humidity qs equals the saturation specific humidity at the sea surface temperature and pressure, q\*(Ts,Ps), is a standard and well-established simplification. This assumption underpins essentially all formulations of latent heat flux in bulk theory (see, e.g., \citep{fairall1996bulk, fairall2003bulk, edson2013exchange}), including the COARE algorithm widely used for air—sea interaction studies. Deviations from saturation at the surface are typically small

relative to other sources of uncertainty in the flux estimation (e.g., wind speed, transfer coefficients).

To clarify this in the manuscript, we have added the following text around line 42

"Following the bulk aerodynamic formulation of surface fluxes, we assume that the water vapor pressure at the ocean surface is at saturation, so that the surface specific humidity \$q\_\mathrm{s}\$ equals the saturation specific humidity \$q\_\*(T\_\mathrm{s}, P\_\mathrm{s})\$. This approximation in bulk theory reflects the near-equilibrium condition at the air--sea interface and is a standard assumption in flux parameterizations, sometimes with a 0.98 correction for typical salinity \citep[e.g.,][]{fairall1996bulk, fairall2003bulk}."

In general, the uncertainty analysis of this paper is underdeveloped. There is instrument error, both from the surface observations of temperature and pressure as well as the lidar observations of cloud base height. There's uncertainty in the analysis framework, including the assumption that surface q is saturated, that the boundary layer is well mixed, etc. A monte carlo estimation of the uncertainty is probably the most straightforward approach here: by randomly perturbing the h, Ts and other variables; assuming that qs is some fraction of q\*; etc., reasonable error bars on the final product can be calculated. As it stands, the value of the product is limited because it is not clear just how trustworthy it is. While the validation relative to some collocated observations is reasonable, that is only for a very specific location and environment. Additional analysis could help greatly inform the utility of this method outside of those regions.

Please see comment 2) in the above cover letter, as well as the new section on scope and limitations, Section 5:

"The method proposed in this study is designed for convective marine boundary layers in which the subcloud layer is well mixed and shallow cumulus clouds are coupled to the surface. These conditions are ubiquitous over the world ocean (Fig.\ref{fig:buoyancy\_flux\_frequency}). Under such conditions, the cloud base height corresponds closely to the lifting condensation level, which depends primarily on the near-surface temperature and humidity.

Despite the ubiquity of favorable conditions, several processes can violate the proxy's assumptions or introduce measurement bias. Cold pools from downdrafts and rain-driven outflows can produce shallow, moist layers decoupled from the overlying

cloud, lowering the observed cloud base relative to the environmental LCL and thus overestimating \(q\_{\mathrm{a}}\). Optically thick or multilayer clouds pose observational limits: a lidar or ceilometer may detect an upper cloud base rather than the lowest, inflating the apparent cloud-base height and biasing the inferred near-surface humidity low; optical-depth and multilayer screening are therefore required. Shallow or weakly mixed layers, in addition to those induced by cold pools, depart from the well-mixed assumption, weakening the cloud-base and surface-humidity link and typically yielding low-biased estimates of \(q\_{\mathrm{a}}\) unless filtered.

Accordingly, the proxy should be applied only where boundary-layer conditions are convective and the detected cloud base is the lowest layer coupled to the surface. These conditions can be diagnosed using coincident lidar backscatter, optical-depth screening, or reanalysis-based buoyancy metrics as in Fig.~\ref{fig:buoyancy\_flux\_frequency}. Within such convective marine regimes, the method's assumptions hold and the resulting estimates of near-surface specific humidity are expected to be reliable."

I also found the order of the paper somewhat difficult to follow. Fig 2 shows the bulk of the work (a relationship between RH and h) that much of the rest of the paper is trying to justify. Fig 4 shows some outliners from Fig 3, but how those outliers are influencing the final q analysis isn't clear. The theory doesn't appear until Sec. 5 when it really should be part of the methodology. The authors should reconsider the odering of the work and optimizing it for a simple flow for the readers.

Thank you. We have reordered the manuscript following this helpful comment to have all theory appearing in a new Section 3, followed by the results.

**Other significant points:**

Regardless of the points raised above, the title of the paper must be changed. As it stands, it sounds like a review of all methods of measuring near-surface specific humidity. A more appropriate title would be "Estimating near-surface specific humidity over the ocean in convective environments from cloud base heights" or something along those lines.

Thank you; this is a great suggestion, and we have changed the title to: "Estimating near-surface specific humidity over convective oceanic regions from cloud base height observations"

I also do not believe this paper falls within the scope of this particular journal. To me, this is much more appropriate as an Atmospheric Measurements Techniques paper as it is directly focused on a measurement technique instead of a more fundamental physical process study. I *strongly* recommend the authors and editor consider a transfer to this sister journal as its scope is far better suited for this particular paper. Some publishers allow reviewers to select "transfer" as an option alongside the standard ones of "reject," "minor revisions," etc. If that option were enabled here at ACP, I would be clicking that box.

Please see 3) Journal fit in the above cover letter.

**Minor points:**

I am not convinced that using the gamma notation over the far more common d/dZ notation improves readability. Since we are dealing with thermodynamic properties, when I see gamma my mind instantly assumes that it is referring to R/cp which made it more challenging to follow the mathematical analysis.

Throughout the manuscript and figures, we have changed this notation to d/dz.

Figure labels have a space between g and kg-1.

Thank you. We removed the space in Figure 8b.

**Reviewer 2:**

This manuscript presents a new method to derive near-surface specific humidity over the ocean based on cloud base height estimates. The authors first introduce their method theoretically, and then make use of EUREC4A ground-based and air-borne data to test the method.

While the manuscript is generally well written and the need of a method for closing this observation gap is well introduced, I have major concerns on the applicability of the method. While the authors claim that their method would be applicable globally e.g. with the help of satellite data, the analysis does not back up this statement as only a limited sub-sample of scenes around Barbados is analyzed. If the main scope of the manuscript is to introduce a novel measurement technique as currently claimed, I recommend re-submitting the manuscript to *AMT* after addressing the major concerns below. If the analysis were tied closer to the shallow convective regime and further analysis were to be added, a publication within ACP could be considered given a major revision of the manuscript.

Please see comments on scope, including a new global buoyancy favorability map (1) and journal fit (3) in the above cover letter.

**Major Comments**

L 14: The authors claim that their results raise the potential for an application to space-borne data, but the analysis does not back up this statement. As EarthCare data has been released, the manuscript should include an analysis of this potential, e.g. in a case study, to back-up the statement.

We believe the contribution of this paper is to introduce a physical idea and its key limitations, and the development of a data product is a logical next step (see point 2) in the above cover letter). More specifically, we discuss the challenges inherent in applying this approach (line 309): "The most promising candidate for obtaining such data is measurements using a spaceborne lidar. Currently the newly-launched EarthCARE satellite \citep{illingworth2015earthcare} provides HSR-Lidar data. It provides backscatter data with a horizontal resolution of about 280\,m and a vertical resolution of 100\,m. While the horizontal resolution appears to be sufficient to apply our method, the limited vertical resolution would imply an error of the near-surface

humidity of about \SI{0.5}{\gram\per\kilo\gram} attributed to the vertical sampling error alone (based on Eq.~\eqref{eq:error} assuming typical values for \$\RH\_\ra, h\$, etc.). With sufficient sampling, and given the variability of cloud base height, it might be possible to obtain greater precision in estimates of the mean cloud base height than what is implied by the single-snapshot vertical resolution. Laser ranging using more sophisticated methods, such as employed by the Global Ecosystem Dynamics Investigation lidar aboard the International Space Station, could provide better estimates of cloud base height. But presently GEDI does not provide data products that allow this capability to be explored. Future satellites refining these technologies could be adapted to the requirements of our proposed method, and potentially could provide coincident estimates of near-surface wind speed and air-sea temperature difference."

L 62-63: This sentence is a central hypothesis for the proposed method, but is not backed by any references. This paragraph needs more information on the applicability and limitations of the proposed method.

Please see responses to the first two comments by reviewer 1.

I am missing information on why the particular flights were chosen for the analysis (route? Conditions? Number of dropsondes?). More flights with the same instrumentation are available from the EUREC4A study (Konow et al, 2021) which should be included in the presented analysis to expand on the statistics and applicability. Performing further analyses in different conditions like observed during the Narval expeditions (Stevens et al, 2019) or, more recently, the PERCUSION campaign (https://orcestra-campaign.org/percusion.html), would further strengthen the understanding of potential limitations and uncertainties of the proposed method.

Starting line 203, we describe why these representative days were chosen as a means of introducing and supporting the development of a physical idea:

"Fig.~\ref{fig:WALES\_cbh\_RH} presents results for two days of the campaign: January 28, 2020 (57 WALES lidar--dropsonde pairings) and February 2, 2020 (32 pairings), which are selected because they sampled a representative range of different cloud base conditions. January 28 was characterized by small cumulus clouds, often referred to as `sugar' clouds \citep[e.g.,][]{stevens2020sugar, bony2020sugar}, while February 2 was characterized by deeper clouds with more stratiform layers near cloud top, often referred to as `flower' clouds \citep[e.g.,][]{stevens2020sugar} and the presence of a strong Saharan dust layer reaching up to 2.5\,km."

LL 286 – 300: The authors suggest that the introduced method offers the potential for a physical retrieval based on space-borne cloud base height estimates. I think that this statement could be further expanded in the analysis, e.g. by adding a designated Section by using available satellite data, and adding a more thorough uncertainty analysis leading to concrete instrument performance parameters that would need to be reached in order to achieve the targeted precision of near-surface humidity estimates.

Please see points (1) and (2) in the cover letter.

Throughout the manuscript, assumptions of central variables are made (e.g. LL 163, 238, 247) but are lacking references. More analysis is needed on the sensitivity of the proposed method to these assumptions.

Line 163: On the backscatter ratio 20, see comment below about L 163.

Line 238: The Justification for "we assume  $\gamma W = 4\% hm - 1$ , a value close to the distribution peak of the cloudy soundings and to the expected theoretical value (Sec. 2)." is the analysis underpinning Figure 7 and is further described in Section 3 on theory and parameter sensitivities.

Regarding the cool skin effect, we have added references for these values 0.1-0.3 K \citep{Fairall1996, Yan2024}.

**Minor Comments**

L 101: how much spatial resolution does the 0.2 second time resolution correspond to depending on flight altitude? We thank the reviewer for this comment. The 0.2-second time resolution corresponds to a horizontal spatial resolution of approximately 42 m at the average flight speed. While the HALO aircraft typically operates between 8 km and 14 km altitude with speeds ranging from 200 m/s to 230 m/s (implying a spatial resolution variation of only about  $\pm 7\%$ ), all analyzed flights in this study were conducted at a nearly constant altitude (~10.4 km) and speed (~210 m/s). We have added the following clarification to the manuscript around line 103:

"The backscatter ratio and aerosol depolarization data are analyzed at 0.2 s time resolution (corresponding to a horizontal spatial resolution of ~42 m for the mean flight speed of 210 m/s) and 7.5 m vertical resolution. For the analyzed flights, the altitude was nearly constant (~10.4 km) and the aircraft speed was about 210 m/s, resulting in consistent spatial resolution across the dataset."

L 163: the sensitivity to the chosen threshold of backscatter ratio of 20 should be further explained. We have added the following lines around line 188: "The threshold

for the backscatter ratio was set to 20 to ensure it lies well above values typically associated with strong aerosol loadings, which can reach up to around 10 for transported desert dust. Lowering the threshold would identify more clouds; however, the resulting change in relative humidity is minimal (about 1\% when varying the threshold from 10 to 20) and becomes negligible (around 0.1\%) when increasing it further from 20 to 40."

Figures: Backscatter ratio 40, left, and backscatter ratio 20, right EUREC4A 02-02-20

And impact on cloud base height statistics for different backscatter (bsr) ratio values:

Fig 2: Outliers are also present in this Figure, at Wa approx. 75% and 88%. As performed later in Fig 3, these outliers should be further addressed. I propose to add the uncertainty of the radiosonde estimates as errorbars to the Figure.

The error bars on the radiosonde estimates are constant and from the manufacturer estimate, hence we do not think that adding that constant value to the figure is necessary.

L 229: the provided range of Bowen ratio and resulting factor is large. A reference and further explanation for the spread should be added to backup the calculations. We have added a reference for the Bowen ratio and focused on the central value of 0.1 given in this study. "The resulting fractional error scales with the Bowen ratio, \$B\$, as \$(1+0.13/B),\varepsilon\_T\$, growing as \$B\$ decreases. For climatological oceanic conditions (\$B\approx0.1\$) \citep{Oliver2005}, this amplification factor is \$\sim2.3\$, which is still insufficient for \$\varepsilon\_T\$ to become comparable to the contribution from an equal \$\varepsilon\_h\$.

L 247: a reference should be provided for the given values 0.1-0.3K. We have added two references: "Typical values for this effect are between \Slrange{0.1}{0.3}{\kelvin} \citep{Fairall1996, Yan2024}." See values in Yan et al., 2024, Fig. 1.

L 304: examples should be provided for "more statistical approaches". Thank you. We have linked back to the examples given in the introduction in this concluding line starting line 323. "At the least, our work suggests that cloud base height information from lidar measurements could be usefully incorporated into reanalyses and into existing statistical flux retrieval frameworks -- such as those used in HOAPS, SeaFlux, IFREMER, or J-OFURO climatologies

\citep[e.g.,][]{gentemann2020fluxsat,liman2018uncertainty,bentamy2017homogenizatio n,tomita2019introduction} -- to better constrain near-surface humidity and surface moisture fluxes."

**Technical Corrections**

L 31-33: define variables Eq (1) in text. We respectfully note that the variables for Eq (1) were defined in the text. We have, however, moved the wind speed definition one line earlier.

L 67: qs,. Extra comma removed.

L 121: unit of q should be gkg-1. Corrected.

L 100: resolution. Typo corrected, thank you.

L 138: incomplete sentence. A missing 'that' is added. ("that are derived from")

L 170: "These two cases" - unclear what this is referring to (introduced later in L 175). We have removed this line since the two cases are introduced below.

L 225 – 230: the sentences need revision with respect to typos and grammar. Thank you. We have edited to be the following, now starting line 155: "This analysis reinforces Eq.~\eqref{eq:error}: the dominant source of uncertainty in \$\Delta q\$ is the estimate of the product of \$h\$ and \$\rd \RH/\rd z\$. Errors in the air—sea temperature contrast, \$\Delta\_a T\$, also propagate into the energy budget via the sensible heat flux, producing same-sign errors in the net surface energy flux. The resulting fractional error scales with the Bowen ratio, \$B\$, as \$(1+0.13/B),\varepsilon\_T\$, growing as \$B\$ decreases. For climatological oceanic conditions (\$B\approx0.1\$) \citep{Oliver2005}, this amplification factor is \$\sim2.3\$, which is still insufficient for \$\varepsilon\_T\$ to become comparable to the contribution from an equal \$\varepsilon h\$."